# Identification of specific Tie2 cleavage sites and therapeutic modulation in experimental sepsis

Temitayo O Idowu[1], Valerie Etzrodt[1], Benjamin Seeliger[2], Patricia Bolanos-Palmieri[1,3], Kristina Thamm[1], Hermann Haller[1], Sascha David[1,4]*

[1]Department of Nephrology and Hypertension, Hannover Medical School, Hannover, Germany; [2]Department of Respiratory Medicine and German Centre of Lung Research (DZL), Hannover Medical School, Hannover, Germany; [3]Department of Nephrology and Hypertension, University Hospital of Erlangen, Erlangen, Germany; [4]Institute for Intensive Care, University Hospital Zurich, Zurich, Switzerland

**Abstract** Endothelial Tie2 signaling plays a pivotal role in vascular barrier maintenance at baseline and after injury. We previously demonstrated that a sharp drop in Tie2 expression observed across various murine models of critical illnesses is associated with increased vascular permeability and mortality. Matrix metalloprotease (MMP)−14-mediated Tie2 ectodomain shedding has recently been recognized as a possible mechanism for Tie2 downregulation in sepsis. Here, we identified the exact MMP14-mediated Tie2 ectodomain cleavage sites and could show that pharmacological MMP14 blockade in experimental murine sepsis exerts barrier protective and anti-inflammatory effects predominantly through the attenuation of Tie2 cleavage to improve survival both in a pre-treatment and rescue approach. Overall, we show that protecting Tie2 shedding might offer a new therapeutic opportunity for the treatment of septic vascular leakage.

*For correspondence:
David.Sascha@mh-hannover.de

Competing interests: The authors declare that no competing interests exist.

## Introduction

Sepsis is a life-threatening clinical syndrome characterized by an overwhelming host's response to an infection leading to organ dysfunction (*Cao et al., 2019*). Its incidence world-wide continues to evolve (*Cao et al., 2019*; *Suarez De La Rica et al., 2016*; *Fleischmann et al., 2016*). Thus far, research on the pathogenesis of sepsis has primarily focused on inflammatory responses which have proven unsuccessful in the translational approach from bench to bedside (*Fisher et al., 1996*). The clinical picture of the syndrome however suggests that the host response in sepsis is characterized by a vascular phenotype with endothelial inflammation, disturbed interaction between the endothelium and the coagulatory system and widespread vascular leakage which directly contributes to morbidity and mortality (*Leligdowicz et al., 2018*). Hence, therapeutic strategies aiming to reduce vascular pathologies might be a treatable part of the injurious host response to ameliorate sepsis morbidity and mortality.

Tie2, a classical transmembrane tyrosine kinase receptor, plays a pivotal role in vascular barrier maintenance (*Parikh, 2017*; *Jongman et al., 2019*). Although attenuation of Tie2 receptor signaling upon infection has been observed for many years (*Parikh, 2017*; *Parikh, 2016*; *Ghosh et al., 2016*), most of the research in the field has focused on manipulating its circulating ligands - the angiopoietins (Angpt) (*David et al., 2012*; *Witzenbichler et al., 2005*). Recent advances have shown that besides its activation, reduced Tie2 expression also directly contributes to pathological permeability (*Ghosh et al., 2016*). In our previous study, Tie2 ectodomain shedding was identified as one of the

possible mechanisms for Tie2 down-regulation upon various inflammatory stimuli. MMP14 was implicated as being the principal Tie2 sheddase (*Thamm et al., 2018*).

We (*Thamm et al., 2018*) and others (*Sung et al., 2011*; *Findley et al., 2007*) have shown that global blockade of MMP activity is sufficient to prevent Tie2 cleavage and thereby vascular leakage in vitro and to improve pulmonary vascular leakage and ultimately, survival in a rat sepsis model in vivo (*Steinberg et al., 2003*). However, the deleterious side effects observed in cancer clinical trials raised severe concerns about its clinical applicability (*Vanlaere and Libert, 2009*; *Coussens et al., 2002*). In contrast, the use of a selective inhibitor for MMP14 might not only help to reduce treatment-associated side effects but might also be an ideal tool to study the mechanistic link between MMP14 and Tie2 shedding in health and disease.

Here, we test the therapeutic potential of selectively inhibiting MMP14 using an affinity matured exosite inhibitor termed E2C6 with the primary aim of preventing endothelial Tie2 shedding in vitro in endothelial cells (ECs) and in vivo in murine sepsis and endotoxemia models. Furthermore, we seek to find the exact MMP14 mediated Tie2 cleavage site(s) to develop potential novel therapeutic strategies in the future. Our study may offer novel insights into the molecular regulation of Tie2 on endothelial barrier stabilization during sepsis.

## Results

### MMP14-dependent Tie2 cleavage occurs at the fibronectin type III domain

First, to better characterize Tie2 ectodomain shedding, we set up to identify the exact site where Tie2 is proteolytically cleaved. We identified a 75–80 kDa Tie2 ectodomain fragment in the supernatant of HUVECs stimulated with TNF-α (*Figure 1A*) similar to that described previously (*Findley et al., 2007*; *Reusch et al., 2001*). Next, we incubated in a cell-free assay recombinant Tie2 protein with the catalytic domain of MMP14 (i.e. cMMP14) and found that cleavage occurs in a concentration-dependent manner (*Figure 1B*). Attempts to immunoprecipitate and isolate the Tie2 protein from these membranes, followed by mass spectrometry (MS) did not reveal any consistent results. We resorted to an in silico analysis to predict the catalytic sites of MMP14 on Tie2. Using the 'cleavepredict' software potential MMP14 cleavage sites of Tie2 could be identified, i.e. $T^{633}$-L, $N^{644}$-I, $S^{648}$-N and $D^{740}$-L. To validate this prediction, we designed two synthetic Tie2 polypeptides spanning the predicted catalytic sites for MMP14 ($^{630}T–^{652}H$ and $^{736}Q–^{745}K$). We then analyzed the cleavage patterns after digestion with the recombinant catalytic domain of MMP14. MS analyses followed by MS/MS fragmentation of selected peptide precursors revealed that one of the polypeptides $^{630}T–^{652}H$ was indeed cleaved at three particular sites $I^{637}$, $N^{644}$ and $S^{648}$ (*Figure 1D*). The second polypeptide remained uncleaved (*Figure 1—figure supplement 1*). We further confirmed functional relevance of these newly identified Tie2 cleavage sites by site-directed mutagenesis analysis. Isoleucine$^{637}$, asparagine$^{644}$ and serine$^{648}$ of Tie2 were mutated to valine, histidine and threonine, respectively. Mutation at I637V and S648T led to a 31% and 32% reduction of Tie2 fragment released into the conditioned media by cMMP14. Meanwhile, mutation at N644H and triple mutation of all three sites led to an 80% and 90% reduction of soluble Tie2 in conditioned media (*Figure 1E*). These results revealed three cleavage sites of Tie2 by MMP14 all within the fibronectin type III domain. In summary, this demonstrates that Tie2 is cleaved by MMP14 possibly at three different sites and targeting these sites might represent a promising therapeutic intervention to treat vascular leakage in sepsis.

### MMP14 blockade attenuates endotoxemia induced Tie2 ectodomain cleavage

To investigate the possible clinical relevance of preventing Tie2 cleavage at these MMP14-mediatied cleavage points, we tested the effects of a specific pharmacological inhibitor of MMP14 (termed E2C6). An ELISA technique was used to detect the extracellular N-terminal epitope of Tie2 under baseline and stimulated conditions (i.e. sTie2). Under baseline conditions, E2C6 was able to reduce spontaneous Tie2 shedding by 60% (*Figure 2A*, bars 1 and 3). Moreover, we observed that pretreatment of HUVECs with 100 nM of E2C6 before TNF-α (50 ng/mL) stimulation led to a decrease in the detectable concentration of sTie2 fragments in cell culture supernatants compared to control-

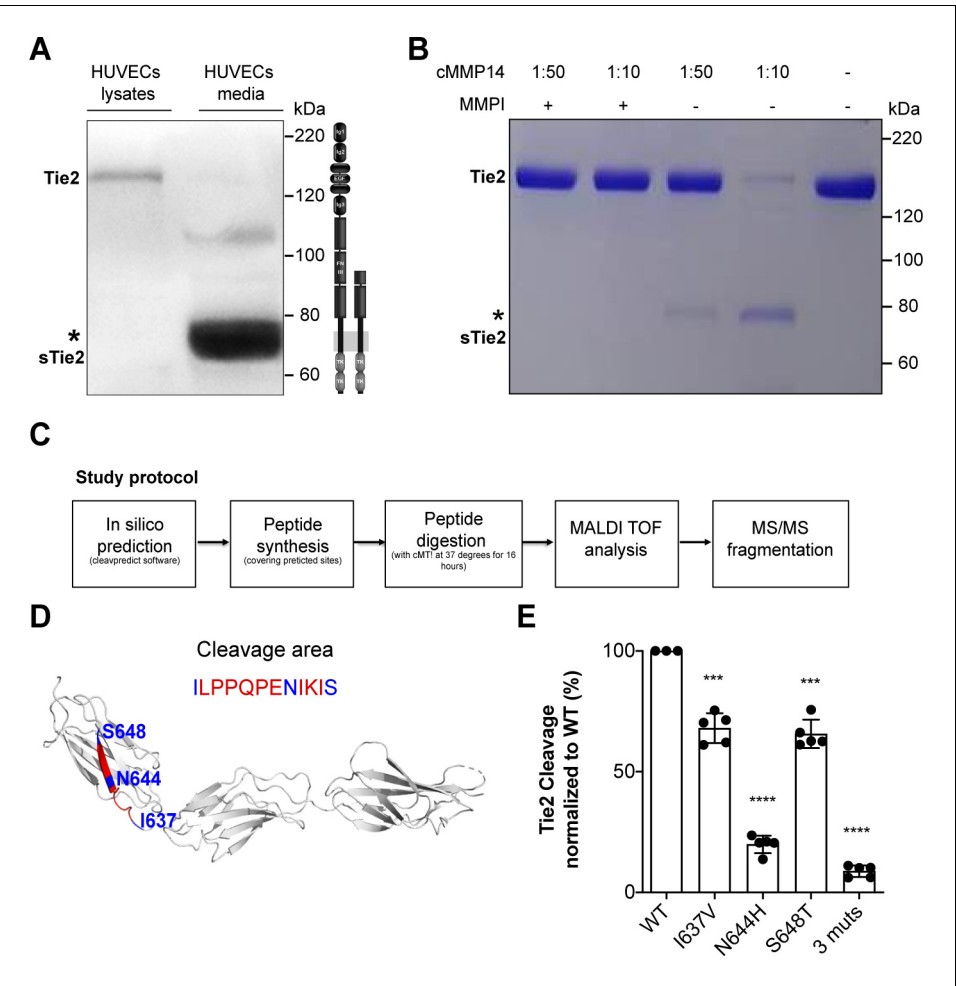

**Figure 1.** MMP14 cleaves Tie2 at the Fibronectin type-III domain on the cell surface. (**A**) Representative Tie2 ectodomain (AB33) immunoblot of the total cell lysates and conditioned media from TNF-α (50 ng/ml) treated HUVECs. The asterisk indicates the truncated 75–80 kDa Tie2 fragments (sTie2). (**B**) Recombinant Tie2 (1 ug) was incubated with the recombinant catalytic domain of MMP14 (termed cMMP14) at two enzyme/substrate ratios (1:50, 1:10, and buffer only) in the presence or absence of GM6001 a general MMP inhibitor (MMPI). The protein mixture was subjected to Coomassie staining. The asterisk indicates the cleaved fragments of Tie2 (sTie2). (**C**) Schematic representation of the study protocol used for the detection of MMP14-mediated Tie2 cleavage sites. (**D**) Crystal structure of Fibronectin type III (FN3) and amino acid sequence of the cleavage area (red) and cleavage sites, I637, N644, S648 (blue) (**E**) Percentage soluble (s)Tie2 in wildtype (WT) or mutant transfected HEK293 cell culture supernatants treated with cMMP14. ***p<0.001, ****p<0.0001 compared with WT in t test (n = 3–5). The online version of this article includes the following figure supplement(s) for figure 1:

**Figure supplement 1.** Recombinant MMP14 cleaves Tie2 polypeptides.

---

treated cells by 85% (**Figure 2A**, bars 2 and 4). Given that E2C6 inhibited Tie2 shedding, we went further to analyze cell lysates for (the remaining) total (t)Tie2 abundance via immunoblot. As expected, reduced sTie2 in the supernatant of the E2C6 group came along with an increased amount of tTie2 within the cell lysate compartment (**Figure 2B–C**). Consistently, endotoxemic mice pretreated with E2C6 were protected from LPS-induced Tie2 cleavage shown by reduced sTie2 in the serum of LPS challenged mice (**Figure 2D**). On the cellular/tissue level LPS-induced loss of tTie2 in the pulmonary vasculature (as a result of shedding) was protected by E2C6 treatment (**Figure 2E–F**). These data suggest that pharmacological inhibition of MMP14 is sufficient to prevent Tie2 proteolytic cleavage both in vitro and in vivo.

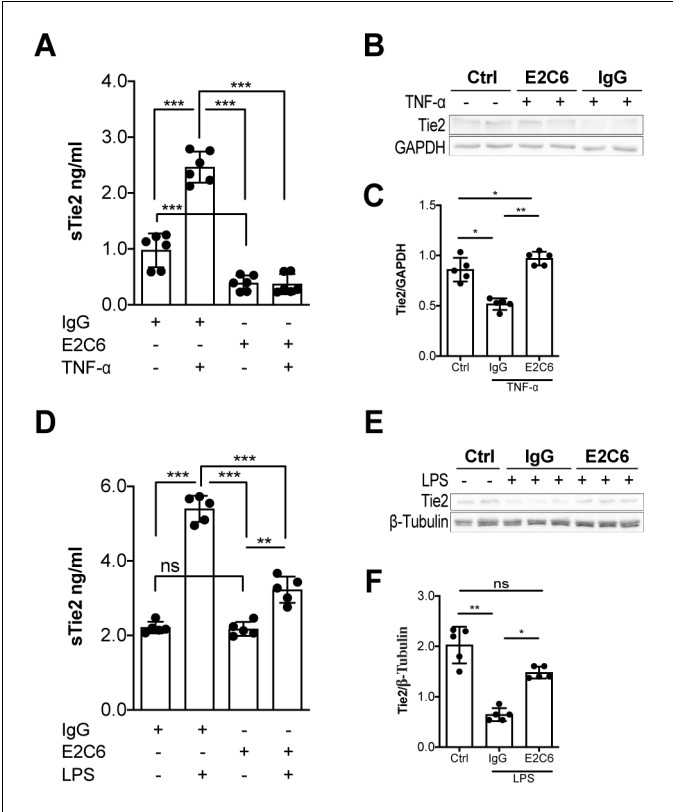

**Figure 2.** MMP14 blockade inhibits Tie2 cleavage both in vitro in HUVECs and in vivo in murine experimental sepsis. Human umbilical vein endothelial cells (HUVECs) pretreated with or without E2C6 (100 nM) for 1 hr were analyzed 24 hr after stimulation with 50 ng/mL TNF-α. (A) sTie2 ELISA of supernatants (**p<0.01, ***p<0.001, n = 6) (B) Representative immunoblot of cell lysates and (C) densitometric quantification of blots (*p<0.05, **p<0.01, n = 5 per group). Mice pretreated with either IgG control or E2C6 for 1 hr were analyzed 16 hr after LPS induced endotoxemia. (D) sTie2 ELISA of murine serum (**p<0.01, ***<0.001, n = 5) (E) Representative immunoblot for Tie2 and beta-tubulin from the murine lung and (F) densitometry quantification of blots (*p<0.05, **p<0.01, n = 5 per group).

## Morphological and functional effects on the septic vasculature upon MMP14 blockade

Next, we analyzed the possible morphological and functional benefits of preventing Tie2 cleavage from the endothelial surface. HUVECs that have been challenged with TNF-α showed a typical phenotype of severe gaps between adjacent cells caused by increased cytoskeletal forces via F-actin polymerization leading to disrupted cell-cell contacts. Again, E2C6 treatment was sufficient to prevent this formation of intercellular gaps compared to control-treated cells (*Figure 3A–B*). Functionally, E2C6 attenuated TNF-α induced increase in the macromolecular passage in an in vitro HRP permeability assay (*Figure 3C*). This finding was confirmed using real-time electrical transendothelial resistance (TER) measurements with TNF-α, and E2C6 treated HUVECs (*Figure 3D*). In vivo permeability was analogously assessed both on the morphological and functional level. We found that E2C6 pretreatment attenuated LPS induced perivascular cuffing - a histological surrogate of lung edema (*Figure 3E–F*). A functional Evans Blue assay quantitatively supported this qualitative observation indicative of a reduced vascular leakage in endotoxemic mice that have been pretreated with E2C6 (*Figure 3G*). These results indicate that MMP14 blockade attenuates Tie2 cleavage thus alleviating endothelial permeability.

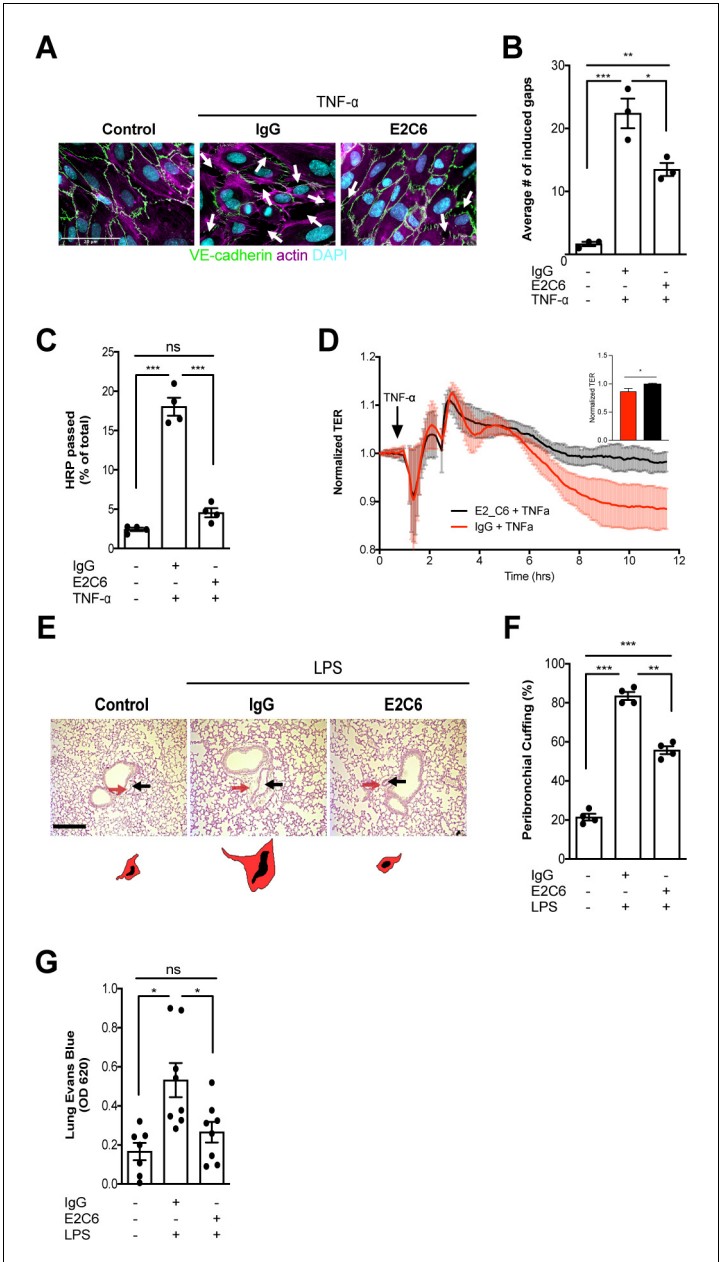

**Figure 3.** MMP14 blockade attenuates sepsis-induced hyperpermeability. (**A**) Representative images of fluorescence immunocytochemistry staining for VE-cadherin (green), F-actin (magenta) and the nuclei (cyan) on confluent HUVECs, pretreated with or without E2C6 (100 nM) for 1 hr, were analyzed 6 hr after stimulation with TNF-α. Arrows (white) indicate inter-endothelial gaps. Scale bar 20 μm. (**B**) Average number (#) of intercellular gaps induced by TNF-α (*p<0.05, **p<0.01, ***p<0.001 in Turkey's multiple comparisons test, n = 3 per group) (**C**) HRP leakage in confluent HUVECs, pretreated with or without E2C6 (100 nM) for 1 hr before TNF-α stimulation (50 ng/ml). (***p<0.001, n = 4) (**D**) Transendothelial Electrical Resistance (TER) was measured using an electrical cell-substrate impedance sensing system (ECIS) in HUVECs that were treated with E2C6 (100 nM) or control IgG and stimulated with 50 ng/ml TNF-α 60 min later. Inset, normalized TER measurement of TNF-α response in E2C6 treated vs control IgG treated cells at the 10 hr mark (*p<0.05, n = 4). (**E**) Periodic acid-Schiff (PAS) staining of paraffin-embedded lung tissue 16 hr after LPS-induced endotoxemia in E2C6 (10 mg/kg) or IgG control antibody (IgG) treatment groups. (n = 5 mice per group). All images show bronchus and their corresponding arteriola (as vasa vasorum of the bronchus surrounded by one common adventitia). The black arrows indicate the vessel area while the red arrows indicate the cuff area. Scale bar 20 μm. (**F**) Semi-quantification of peribronchial cuffing was performed by surveying whole lung sections. (***p<0.001, n = 4 per group). (**G**) In vivo pulmonary Evans blue

*Figure 3 continued on next page*

*Figure 3 continued*

permeability assay in mice pretreated with 10 mg/kg IgG control or E2C6 1 hr before LPS-induced endotoxemia (*p<0.05, n = 7–8 per group).

## MMP14 blockade inhibits endothelial inflammation and tissue infiltration

Given that Tie2 activation controls not only permeability but also endothelial inflammation via adhesion molecule suppression (*van Meurs et al., 2009*; *Hughes et al., 2003*; *Kim et al., 2001*), we tested if protection from Tie2 cleavage might analogously exert beneficial properties. We found that the increase of both ICAM-1 and VCAM-1 mRNA could be ameliorated (but not prevented) if E2C6 treatment was applied to HUVECs upon TNF-α stimulation (*Figure 4A–C*) or mice challenged with LPS (*Figure 4D–F*). Of note, there was no effect on E-selectin expression in both settings. Physiologically, ICAM-1 and VCAM-1 are involved in the transmigration of neutrophils into the tissue by progressing the rolling process towards a firmer adhesion. Consistently, MMP14 blockade profoundly reduced pulmonary Gr-1 neutrophil infiltration in LPS-induced endotoxemia in mice (*Figure 4G–H*). From these results, we inferred that MMP14 blockade prevents Tie2 shedding, which in turn might protect the endothelium against systemic endothelial inflammation and pathological systemic neutrophil recruitment. Reduced tissue infiltration of inflammatory cells further suggests that MMP14 blockade might also exert an additional protective effect by dampening local cytokine release in all organs. To test this, we analyzed a panel of serum levels in endotoxemic mice treated with E2C6 or control Ab. Indeed, interferon (IFN) γ, interleukin (IL)−6, macrophage chemoattractant protein (MCP)−1, and TNF-α were significantly reduced in E2C6 treated group compared to the IgG treated control group (*Figure 5A–D*). A similar trend was observed in pulmonary IL-6 and TNF-α transcript (*Figure 5—figure supplement 1*). Interestingly, the anti-inflammatory cytokine IL-10 was unchanged (*Figure 5E*).

## Effect of MMP14 blockade on clinical outcomes in experimental sepsis

To test whether the anti-permeability and anti-inflammatory properties observed by E2C6 in endotoxemic mice would translate to an improved outcome, we analyzed Kaplan Meier survival in a clinically meaningful polymicrobial sepsis model (i.e. cecal ligation and puncture (CLP)). First, mice were treated i.p. with either E2C6 or control IgG one hour before CLP surgery. To semi-quantify the severity of the disease, we daily scored the morbidity in a blinded fashion (*Table 1*) and found a better performance of the E2C6 treated animals over the observation time of 96 hr (*Figure 6A*). Only 18% of mice pretreated with control IgG survived compared with 56% E2C6 groups (*Figure 6C*, p=0.009). Lastly, in a therapeutic scenario, we tested whether the delayed administration of E2C6 would also affect survival. Therefore, mice were subjected to CLP first and then treated with E2C6 i.p. at 2, 24, and 48 hr after surgery. Also in this therapeutic approach, E2C6 was sufficient to reduce disease severity (*Figure 6B*) and to improve survival by 33% in an otherwise 100% lethal CLP model (*Figure 6D*, p=0.03).

## Discussion

Here we show for the first time that Tie2 is cleaved at positions I637, N644 and S648 within the fibronectin type III (FN3) domain by MMP14, resulting in the production of a 75–80 kDa soluble receptor fragment (termed sTie2). Most importantly, site-directed mutagenesis revealed that a single N644H mutation is sufficient to counteract most of the Tie2 cleavage paving the road towards novel, innovative treatment approaches. As a proof of principle, we used an affinity matured exosite antibody termed E2C6 against MMP14 and found that it is sufficient to completely block the injurious process of Tie2 ectodomain shedding. Functionally, E2C6 ameliorated endothelial vascular leakage and inflammation in vitro and in vivo and even improved hard outcome in murine experimental sepsis.

Our results further support the concept by *Ghosh et al., 2016* that intact Tie2 expression (asides from activation) is necessary for canonical downstream signaling and thereby for barrier defense. Further in this regard, a recent study by *Braun et al., 2020* reported that endothelial Tie2 activation

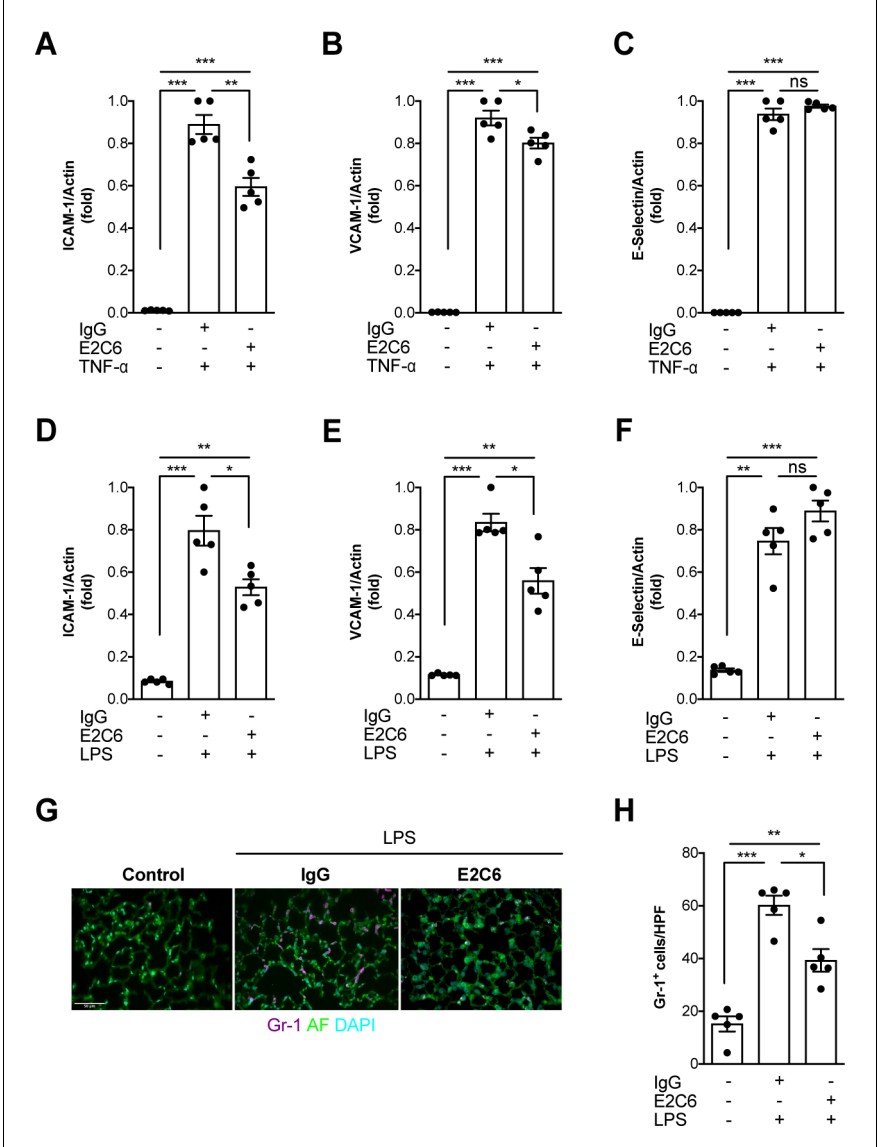

**Figure 4.** MMP14 blockade inhibits inflammation and neutrophil adhesion. Messenger (m) RNA expression of (**A**) ICAM-1, (**B**) VCAM-1, and (**C**) E-selectin from HUVECs pretreated with either E2C6 or control IgG before TNF-α stimulation for 24 hr (*p<0.05, **p<0.01, ***p<0.001, n = 5 per group, ns = not significant). Lung mRNA 16 hr after LPS induced endotoxemia for markers of vascular inflammation (**D**) ICAM-1, (**E**) VCAM-1, (**F**) E-selectin. (*p<0.05, **p<0.01, ***p<0.001, n = 5 per group, ns = not significant,) (**G**) Representative lung immunostaining for granulocyte differentiation antigen (Gr)−1 (red) was performed 16 hr after cecal ligation puncture (CLP) or sham surgery in E2C6 (10 mg/kg) or IgG control antibody (IgG)-treated mice (nuclear staining with 4',6-diamidino-2-phenylindole, (blue), autofluorescence is shown in green). (n = 5) Scale bar 50 μm. (**H**) Semiquantification of whole lung cross-sections by evaluating Gr-1+ cells per high power field (HPF) (HPF = 40 × magnification) (*p<0.05, **p<0.01, ***p<0.001, n = 5 per group).

limits diapedeses induced leakage. The cleavage of the Tie2 ectodomain prevents protective Angiopoietin-1-Tie2 ligation and the cleaved Tie2 receptor can even function as a ligand trap in the circulation further inhibiting protective Angpt-1 activity (*Alawo et al., 2017*; *Zhang et al., 2019*). Besides these effects on Tie2 activation by ligands, we found that the inhibition of shedding to protect Tie2 surface expression is sufficient to ameliorate sepsis-induced vascular barrier breakdown.

One can speculate that MMP14 inhibition with E2C6 will affect canonical Tie2 downstream signals other than permeability. And indeed, we have observed additional anti-inflammatory effects as

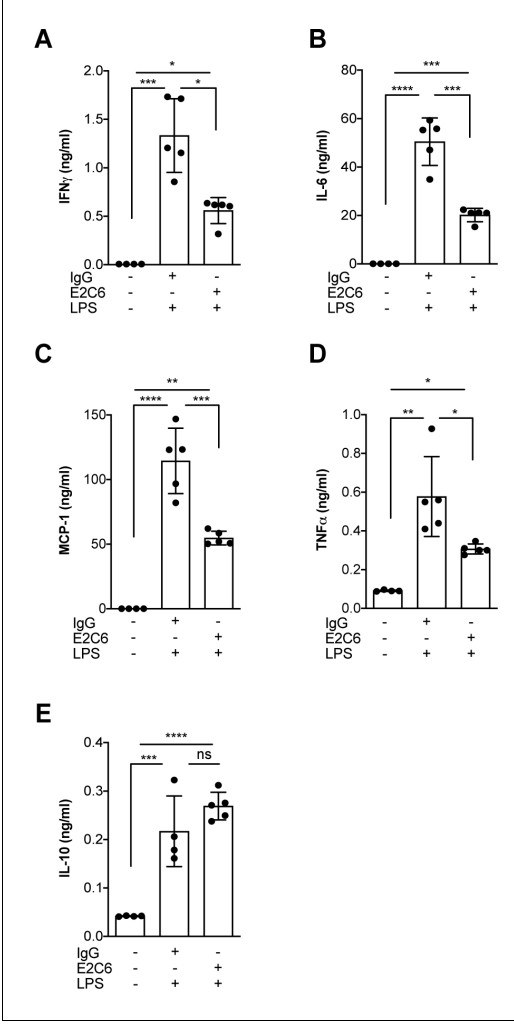

**Figure 5.** MMP14 blockade inhibits endotoxemia-induced pro-inflammatory cytokine release. Quantification of circulating levels of (A) IFNγ, (B) IL-6, (C) IL-1β, (D) TNF-α and (E) IL10 in the serum of mice pretreated with E2C6 (10 mg/kg) or control antibodies for 1 hr, followed by LPS for 16 hr (*p<0.05, **p<0.01, ***p<0.001, ****p<0.0001, n = 5 per group, ns = not significant). Data are the mean ± SD.

The online version of this article includes the following figure supplement(s) for figure 5:

**Figure supplement 1.** E2C6 attenuates LPS induced pulmonary proinflammatory cytokine transcription.

shown by a decrease in endothelial adhesion molecule expression, (*Kim et al., 2001*) neutrophil influx and local cytokine production. Asides from Tie2 signaling, *Kaneko et al., 2016* showed that MMP14 blockade can directly modulate macrophage cytokine production in vitro, as well as promote changes in macrophage polarization favoring an anti-inflammatory phenotype. It is also noteworthy that besides the global decrease in LPS-induced pro-inflammatory cytokines upon MMP14 blockade, a slight increase was observed in the anti-inflammatory IL-10 cytokine in our study. This observation underlines potential off-target anti-inflammatory properties possibly explained by phenomena such as macrophage polarization.

By preventing Tie2 cleavage together with anti-inflammatory properties, it was of no surprise that E2C6 treated mice subjected to CLP had a better survival both in the pretreatment and rescue scenario compared to the IgG control group. These findings underline that MMP14 blockade might be an interesting therapeutic target against a key component of the hosts' response to infection.

Based on the newly identified Tie2 cleavage sites, a possible future approach for the treatment of capillary leakage in sepsis might be to develop an antibody or small molecule that specifically binds with high affinity to the Tie2 cleavage sites hence preventing MMP14 mediated cleavage.

Overall the current study has some limitations; first, our findings on the identified Tie2 cleavage sites and the site-directed mutation of these sites were based on in silico and in vitro studies alone, it might be necessary to validate our finding in vivo. Also, we did not show whether the effect of MMP14 blockade is systemic or organ-specific; however, given the broad expression of Tie2 in various endothelial cells and the presumably broad effect of MMP14 on Tie2 shedding, we would predict that MMP14 blockade has a systemic rather than organ-specific effect. This is supported by the observation that soluble Tie2 (sTie2) in the serum of endotoxemic mice compared to control mice is increased accompanied by a decrease in total Tie2 (tTie2) abundance in lung homogenates. We decided to focus on the lungs because of its hallmark clinical role in sepsis together with the known high abundance of endothelial cells per organ tissue. Furthermore, we did not show whether MMP14 blockade regulates other endothelial and non-endothelial proteins to exert its protective effect. Further experiments might be required to identify possible off-target regulations upon E2C6 treatment. Lastly, given that MMP14 does not work in isolation but rather in complex enzymatic cascades triggering the activation of other enzymes, there is also a risk for adverse events if used in human trials. However, considering the timing aspect for a sepsis therapeutic compared to an adjunctive in an oncology trial, this risk might be of minor relevance.

**Table 1.** Activity score to evaluate severity of illness in septic mice.

| Score | Activity | General condition | Behaviour |
|---|---|---|---|
| 1 | Very active | Smooth fur, clear eyes, clean orifices (body openings) | Vigilant, curious, normal movements |
| 2 | Active | Smooth fur, clear eyes, clean orifices | Vigilant, normal movements |
| 3 | Less active | Matted fur, fur defects, eyes not completely open | Vigilant, quiet, reduced movements, normal posture, reduced corporal hygiene |
| 4 | Restricted | Dull fur, standing fur, eyes not completely open, unkempt orifices | Quiet, frequent persistence, restricted reaction on environmental stimuli, restricted body care |
| 5 | Apathetic | Dirty, dull fur, eyes closed, clogged or humid orifices, crooked posture | Self-isolation, no significant activity |
| 6 | Moribund (Death is expected) | Eyes closed, lateral position, shallow breathing, cramps, cold animal | No activity, no reaction on environmental stimuli |

In conclusion, for the first time we demonstrated that Tie2 is indeed cleaved at three distinct sites within the extracellular fibronectin type III domain leading to 75–80 kDa soluble receptor fragments (sTie2). Moreover, this study is the first to suggest that pharmacological MMP14 inhibition in sepsis exerts barrier protective effects predominantly through the inhibition of the injurious Tie2 cleavage at these ectodomain sites.

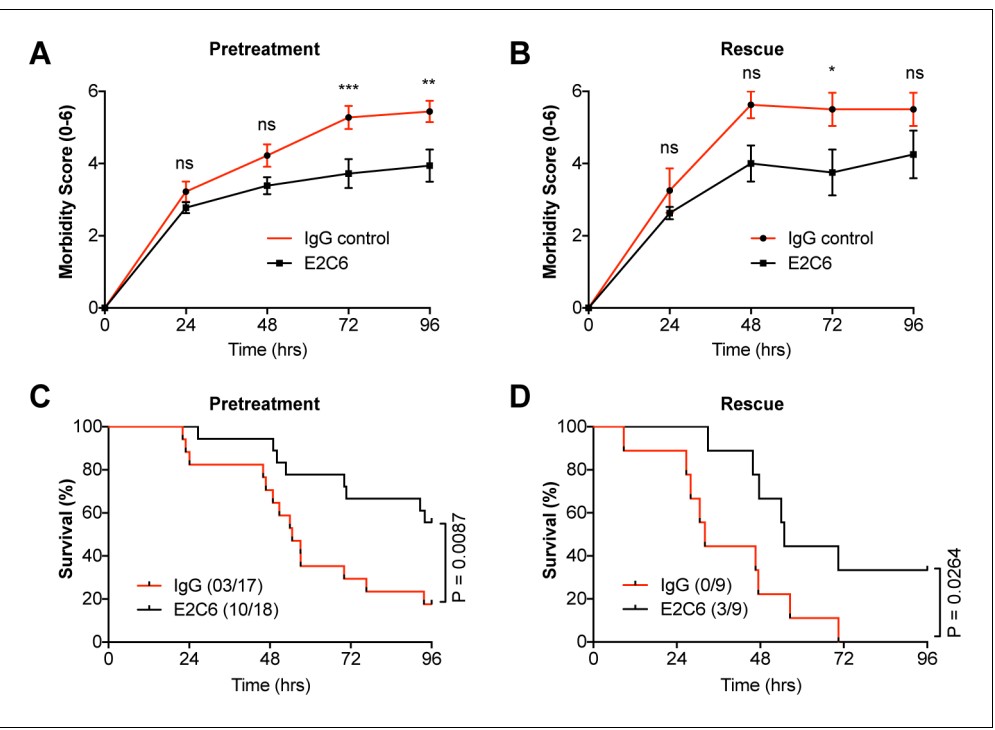

**Figure 6.** MMP14 blockade improves survival in experimental sepsis. The morbidity of sepsis and severity of illness was semi-quantitatively assessed by an in-house scoring system (activity score, *Table 1*) (**A**) mice pretreated with either E2C6 (10 mg/kg) or control IgG intraperitoneally for 1 hr before CLP (n = 17–18 per group) (**B**) Mice subjected to CLP first and then treated with 10 mg/kg of E2C6 at 2, 24, and 48 hr after CLP. (n = 9 per group) (*p<0.05, **p<0.01, ***p<0.001, ns = non significant, in Bonferroni posttest of 2-way ANOVA). Kaplan-Meier survival analysis after CLP-induced sepsis in (**C**) mice pretreated with either E2C6 (10 mg/kg) or Control IgG intraperitoneally for 1 hr prior to CLP (**D**) Mice subjected to CLP first and then treated with 10 mg/kg of E2C6 at 2, 24, and 48 hr after CLP. Number in parentheses represents the number of surviving mice per each group. Statistical significance was analyzed by a log-rank test.

# Materials and methods

## Key resources table

| Reagent type (species) or resource | Designation | Source or reference | Identifiers | Additional information |
|---|---|---|---|---|
| Antibody | Tie2 (C-20) (Rabbit polycloned IgG) | Santa Cruz | Cat#: SC-324 | WB: 1:1000 |
| Antibody | GAPDH (Rabbit polycloned IgG) | Santa Cruz | Cat#: SC-25778 | WB: 1:1000 |
| Antibody | β-Tubulin (Rabbit polycloned IgG) | Santa Cruz | Cat#: SC-9104 | WB: 1:1000 |
| Antibody | RAT anti Mouse Ly-6B.2 ALLOANTIGEN | Bio-Rad | Cat#: MCA771G | IF:1:200 |
| Antibody | CD144 (VE-cadherin) | BD Biosciences | Cat#: 555661 | IF:1:100 |
| Antibody | Human IgG2 isotype control | BioXcell | Cat# BE0301 | |
| Antibody | Anti-MMP14 (E2C6) | *Botkjaer et al., 2016* | PMID:26934448 | A kind gift from Prof. Gillian Murphy and Dr. Yoshifumi Itoh |
| Commercial assay or kit | Human Tie-2 DuoSet | R and D | Cat#: DY5159 | |
| Commercial assay or kit | Mouse Tie-2 Quantikine ELISA Kit | R and D | Cat#: MTE200 | |
| Commercial assay or kit | CBA Mouse Inflammation Kit | BD | Cat#: 552364 | |
| Peptide, recombinant protein | Recombinant Human TNF-alpha | R and D | Cat#: 210-TA-020 | |
| Peptide, recombinant protein | Recombinant Human MMP14 (catalytic domain) | Enzo | Cat#: ALX-201–098 C010 | |
| Peptide, recombinant protein | Recombinant Human TIE2 protein | Sino Biological | Cat#: 10700-H03H | |
| Software, algorithm | ImageJ | NIH, Bethesda, MD (Version 1.52P) | | https://imagej.nih.gov/ij/ |
| Software, algorithm | Prism8 | Graphpad | | https://www.graphpad.com/scientific-software/prism/ |
| Software, algorithm | Flowjo | BD | | https://www.flowjo.com/solutions/flowjo/downloads |
| Other | LPS *Escherichia coli* serotype O111:B4 | Sigma-Aldrich | Cat#: L4130 | |

## Cell culture, stimulation, transfections, antibodies, DNA constructs, and reagents

HEK293T cells were purchased while Human umbilical vein endothelial cells (HUVECs) were isolated from human umbilical veins upon informed consent from donors and approval from the ethical

committee of Hannover Medical School (Nr. 1303–2012). All cells were mycoplasma contamination free. For stimulation experiments, HUVECs were first pretreated with 100 nM E2C6 or control for 1 hr followed by TNF-α (50 ng/ml) stimulation. HEK293T cells were transfected with expression vectors encoding full-length or mutated human Tie2 (SINO Bio) using X-treme Gene HP DNA transfection reagent (Roche) according to the manufacturer's instructions. Based on a cleavpredict (*Kumar et al., 2015*) in silico and mass spectrometry analysis, the predicted MMP14-mediated Tie2 cleavage sites were mutated. Site-directed-mutagenesis of identified sites was carried out at Eurofins Genomics (Ebersberg, Germany) and confirmed by Sanger sequencing (Microsynth seqlab GmbH). Unless otherwise specified, all chemicals and reagents used were purchased from Sigma-Aldrich (St. Louis, MO, USA).

## Immunoblotting

Cells were washed with cold PBS before being homogenized in radioimmunoprecipitation assay buffer (RIPA) followed by centrifugation at 4°C for 15 min at 12000 rpm. Protein concentration present in the supernatant was determined with the Pierce BCA Protein Assay Kit (Thermo Scientific, Rockland, IL). Proteins were resolved with a 10% polyacrylamide gel electrophoresis then transferred to PVDF (polyvinylidene fluoride) membranes (Merck Millipore, Darmstadt, Germany). The membrane was blocked with 3% bovine serum albumin (BSA) and incubated with a primary antibody overnight (4°C) followed by incubation with 2nd antibody for one hour at room temperature. Bands were visualized with SuperSignal West Pico Chemiluminescent Substrate (Life Technologies) and Versa Doc Imaging System Modell 3000 (BioRad). Quantification of immunoblots was done using ImageJ.

## Real-time quantitative PCR

Total RNA was extracted using the RNeasy Mini/Micro Kit (Qiagen, Hilden, Germany) followed by reverse-transcription using Transcriptor First Strand cDNA Synthesis (Roche Diagnostics). RT-qPCR analysis was performed using a LightCycler 480 II (Roche). Gene expression was normalized to the expression of the housekeeping gene, yielding the ΔCT value.

## In vitro and in silico MMP14 cleavage assay

The recombinant catalytic domain of MMP14 (ALX-201–098 C010) and recombinant Tie2 (10700-H03H) were purchased from Enzo and Sino biologicals, respectively. The recombinant Tie2 consists of ECD human Tie2 protein (1–745 amino acids) fused with HIS and Fc-tag at the amino terminus. Mixtures were incubated in the assay buffer (50 mM Tris-HCl pH 7.5, 150 mM NaCl, 5 mM $CaCl2\%$ and 0.025% Brij35) at 37°C for 16 hr. The protein mixture was subjected to western blotting and Coomassie staining analyses. MMP14-mediated Tie2 cleavage sites were identified using cleavpredict (*Kumar et al., 2015*). Molecular structure analysis and visualization of the cleavage sites were done using PyMOL software using the crystal structure of the membrane-proximal three fibronectin type III domains of Tie2 (FN3) deposited in the PDB database (PDB ID: 5UTK) (*Moore et al., 2017*).

## Immunofluorescence staining for cultured ECs

HUVECs were grown to confluency on glass coverslips pre-coated with collagen. Cells were pretreated with 100 nM E2C6 or control antibody for 1 hr followed by TNF-α (50 ng/ml) stimulation for 6 hr. At the end of the experiment, Cells were washed twice with PBS and fixed with 4% formaldehyde, permeabilized with 0.1% Triton X-100 in PBS and blocked with 10% donkey serum (Jackson Immuno Research Inc, West Grove, PA, USA). Afterwards, Coverslips were incubated with primary antibodies for one hour followed by incubation with secondary antibodies for an additional one hour. Staining of the nucleus was carried out for 10 s using 4′,6-diamidino-2-phenylindole (DAPI) purchased from Sigma-Aldrich (St. Louis, MO). The coverslips were affixed with aqua-poly/mount (Polysciences Inc, Eppelheim, Germany) and images were taken with a Leica DMI 6000B microscope with the same gain, exposure, and offset settings across all treatment conditions.

### Transendothelial Electrical Resistance (TER)

TER was measured using an electric cell-substrate impedance sensing system (ECIS) (Applied Bio-Physics Inc) (David et al., 2011a). Values were pooled at discrete time points and plotted versus time. The normalized TER was derived by dividing each condition's endpoint resistance by its starting resistance.

### Mass spectrometry

The polypeptides of human Tie2 (630T–652H and 736Q–745K) were synthesized at GL Biochem (Shanghai China) and incubated with the recombinant catalytic domain of MMP14 in the assay buffer (50 mM Tris-HCl pH 7.5, 150 mM NaCl and 5 mM CaCl2) at 37°C for 16 hr. MALDI-TOF-MS and MS/MS fragmentation of selected peptide precursors were carried out at TOPLAB GmbH, Germany.

### Fluorescent immunohistochemistry

Paraffin-embedded sections (1.5 μm) from lungs were labelled with primary antibody against Gr-1 (AbD serotec, Puchheim, Germany). Followed by incubation with secondary antibodies, we used goat anti-rat IgG-HRP (Santa Cruz Biotechnology, CA, USA). For global histomorphologic analysis of lungs, we used periodic acid-Schiff (PAS) staining.

### Transwell permeability assay

For measurement of HRP passage, confluent HUVEC cells were seeded in transwell inserts with 0.4 μm polycarbonate membranes in a 24-well plate (Corning) and grown to confluence in 5 days as previously described (Chen and Yeh, 2017). Cells were pretreated with either 100 nM E2C6 or control antibody for 60 min before stimulation with TNF-α (50 ng/ml) for 12 hr. Medium in the top chamber was changed to streptavidin-HRP-containing medium and incubated in a 37°C incubator for an additional 30 min. Samples were taken from the lower compartment. The HRP concentration was detected by measuring the absorption at 450 nm after addition of TMB substrate.

### Cecal Ligation and Puncture (CLP) sepsis model and treatment regimen

Male C57bl/J6 mice, 10–12 weeks of age, were anaesthetized with inhaled isoflurane and subjected to CLP surgery by a single operator, as previously described (David et al., 2012; Wen, 2013). For survival studies in a pre-treatment setting, mice were given a single intraperitoneal injection of 10 mg/kg of control IgG or Anti-MMP14 (E2C6) one hour before CLP surgery. To test whether delayed administration of Anti-MMP14 (E2C6) improves survival in a rescue scenario, mice were subjected to CLP first and then treated with 10 mg/kg of control IgG or Anti-MMP14 (E2C6) i.p. at 2, 24, and 48 hr after surgery. The survival rate of mice was determined for four days.

### LPS-endotoxemia model and treatment regimen

Male C57bl/J6 mice,10–12 weeks of age were given a single intraperitoneal injection of 10 mg/kg of control IgG or E2C6 one hour before the administration of 17.5 mg/Kg of body weight of LPS from E. coli serotype O111: B4 (Sigma–Aldrich) intraperitoneally (i.p.). After 16 hr, the mice were sacrificed for organ and blood harvest for further molecular analysis.

### Evans blue permeability assay

Sixteen hours after LPS administration, animals were anaesthetized with inhaled isoflurane. 2% wt/vol Evans blue (100 μl) was injected into the retro-orbital sinus. At exactly 10 min after Evans blue injection, mice were euthanized and perfused with 10 ml of PBS 2 mM EDTA for 5 min through a cannula placed in the right ventricle, after which organs were harvested and homogenized in formamide for extraction and measurement of Evans blue as previously described (David et al., 2011b).

### Statistical analysis

Results are presented as mean ± SEM unless otherwise specified. A p-value less than 0.05 were considered statistically significant. Means were compared by unpaired t test, and for groups of three or more conditions, ANOVA with post-hoc Turkey's test was used. Survival data were analyzed by log-rank test and visualized by Kaplan-Meier curves. Analysis and graphical presentation were performed using GraphPad Prism eight software.

## Acknowledgements

We thank members of SD laboratory, most especially Yvonne Nicolai, for excellent technical assistance. Also, we thank Yoshifumi Itoh and Gillian Murphy for the Anti-MMP14 antibody (E2C6).

## Additional information

### Funding

| Funder | Grant reference number | Author |
|---|---|---|
| Deutsche Forschungsgemeinschaft | DA1209/4-3 | Sascha David |

The funders had no role in study design, data collection and interpretation, or the decision to submit the work for publication.

### Author contributions

Temitayo O Idowu, Conceptualization, Data curation, Formal analysis, Investigation, Visualization, Methodology, Writing - original draft, Writing - review and editing; Valerie Etzrodt, Data curation, Writing - review and editing; Benjamin Seeliger, Resources, Writing - review and editing; Patricia Bolanos-Palmieri, Methodology, Writing - review and editing; Kristina Thamm, Conceptualization, Methodology, Writing - review and editing; Hermann Haller, Writing - review and editing; Sascha David, Conceptualization, Resources, Data curation, Supervision, Funding acquisition, Methodology, Project administration, Writing - review and editing

### Author ORCIDs

Temitayo O Idowu (iD) https://orcid.org/0000-0002-3321-8589
Sascha David (iD) https://orcid.org/0000-0002-8231-0461

### Ethics

Animal experimentation: All animal experiments were approved by the local committee for care and use of laboratory Animals (LAVES Lower Saxony, reference No. AZ/17-2499) and were performed according to international guidelines on animal experimentation.

### Decision letter and Author response

Decision letter https://doi.org/10.7554/eLife.59520.sa1
Author response https://doi.org/10.7554/eLife.59520.sa2

## Additional files

### Supplementary files
• Transparent reporting form

### Data availability

All data generated or analysed during this study are included in the manuscript and supporting files.

The following previously published dataset was used:

| Author(s) | Year | Dataset title | Dataset URL | Database and Identifier |
|---|---|---|---|---|
| Moore JO, Lemmon MA, Ferguson KM | 2017 | Crystal structure of the membrane proximal three fibronectin type III (FNIII) domains of Tie2 (Tie2[FNIIIa-c]) | https://www.rcsb.org/structure/5UTK | RCSB Protein Data Bank, 5UTK |

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
