## [Decision Letter]

**Acceptance summary:**

The authors have identified the site of Tie2 that is cleaved by MMP14 by which Tie2 is shed. They demonstrate that attenuation of this shedding of Tie2 is associated with protective effects in a murine sepsis model. This study contributes to the knowledge of endothelial function and the role of Tie2 in sepsis and opens up novel strategies to intervene with endothelial dysfunction in sepsis.

**Decision letter after peer review:**

Thank you for submitting your article "Identification of specific Tie2 cleavage sites and therapeutic modulation in experimental sepsis" for consideration by *eLife*. Your article has been reviewed by three peer reviewers, including Frank L van de Veerdonk as the Reviewing Editor and Reviewer #1, and the evaluation has been overseen by Matthias Barton as the Senior Editor. The following individual involved in review of your submission has agreed to reveal their identity: Jaap D Van Buul (Reviewer #2).

The reviewers have discussed the reviews with one another and the Reviewing Editor has drafted this decision to help you prepare a revised submission.

Summary:

The authors show that downregulation of Tie2 in endothelial cells is an important feature in in sepsis and have further explored the mechanisms leading to Tie2 downregulation in a murine model and in vitro. The authors show that this is MMP14 dependent and continue to show that blocking MMP14-mediated Tie2 shedding protects the loss of vascular barrier and has anti-inflammatory effects using experimental murine sepsis. This work may provide a new therapeutic target for vascular leakage during sepsis. The work is novel, straightforward and the data support the conclusions.

Essential revisions:

This is in potential a very strong measure and the work is novel, and done thoroughly. The conclusions drawn make sense and the experiments carried out are of high quality and include the proper controls. I have some comments that I wish to see being addressed by the authors.

1) Tie 2 shedding would also reduce inflammation. How then? Does it directly target NF-kB activation?

2) They report that the MMP14 inhibition works in the lungs. What about other organs? In other words, is the effect systemic or is there any organ specificity? Can they discuss this in the Discussion section, or perhaps have data to include?

3) As the inhibition of tie2 cleavage results in the prevention of (LPS-induced) loss of resistance/permeability, did they carefully check junction protein expression, e.g. VE-cadherin but in particular the tight junction proteins, known to control permeability in full-grown Ec monolayers (e.g. claudins, occludins).

4) Recently, a study by Braun et al., Blood 2020, from the group of Vestweber was published and reported on the fact that Tie2 activation limits the leakage that is induced by transmigrating leukocytes. This study would be complementary to theirs and should be cited in the Discussion section.

5) How sure can you be there are no cleavage sites outside of the synthetic Tie2 sites you generated? I am not familiar with the reliability of the software used, but apparently it at least wrongly predicted a cleavage site in D740-L. If the authors aren't sure these are the only three cleavage sites, perhaps the conclusion should be changed to leave open the possibility that MMP14 cleaves elsewhere (subsection “MMP14-dependent Tie2 cleavage occurs at the Fibronectin type III domain”).

6) In the in vitro experiments, TNFa was used to induce inflammation, whereas in the in vivo experiments LPS was used. I understand that LPS is not as effective as TNFa in HUVEC, but perhaps the authors could explain possible other reasons for this.

7) In Figure 3A, the holes between cells look really large. Is this image a representative image of all experiments? If so, maybe the authors can think of a way to quantify average hole size and/or number of holes induced by TNF-α.

---

## [Author Response]

Essential revisions:This is in potential a very strong measure and the work is novel, and done thoroughly. The conclusions drawn make sense and the experiments carried out are of high quality and include the proper controls. I have some comments that I wish to see being addressed by the authors.1) Tie 2 shedding would also reduce inflammation. How then? Does it directly target NF-kB activation?

We thank the reviewer for his/her thoughtful question. In our setting and in the experiments that we performed, Tie2 shedding did not relevantly reduce inflammation. That being said we also didn’t focus on this but rather on the regulation of the vascular barrier. Nevertheless, others have shown that Tie2 activation is sufficient to inhibit the expression of surface adhesion molecules ICAM-1 and VCAM-1 (1) (all of which are downstream of the NF-κB pathway) by recruiting the adaptor protein growth factor receptor-bound protein 2 (ABIN2) (2).

2) They report that the MMP14 inhibition works in the lungs. What about other organs? In other words, is the effect systemic or is there any organ specificity? Can they discuss this in the Discussion section, or perhaps have data to include?

We agree with the reviewer. Given the broad expression of Tie2 in various endothelial cells and the presumably broad effect of MMP14 on Tie2 shedding, we would predict that MMP14 blockade has a systemic rather than organ-specific effect. This is supported by the observation that soluble Tie2 (sTie2) in the serum of endotoxemic mice compared to control mice is increased (Figure 2D) accompanied by a decrease in total Tie2 (tTie2) abundance in kidney (Author response image 1) and lung homogenates (Figure 2E). We decided to focus on the lungs because of its hallmark clinical role in sepsis together with the known high abundance of endothelial cells per organ tissue. As suggested, this has been included in the Discussion section.

Discussion:

“Also, we did not show whether the effect of MMP14 blockade is systemic or organ-specific; however, given the broad expression of Tie2 in various endothelial cells and the presumably broad effect of MMP14 on Tie2 shedding, we would predict that MMP14 blockade has a systemic rather than organ-specific effect. […] We decided to focus on the lungs because of its hallmark clinical role in sepsis together with the known high abundance of endothelial cells per organ tissue.”

**Author response image 1. sa2fig1:** 

3) As the inhibition of tie2 cleavage results in the prevention of (LPS-induced) loss of resistance/permeability, did they carefully check junction protein expression, e.g. VE-cadherin but in particular the tight junction proteins, known to control permeability in full-grown Ec monolayers (e.g. claudins, occludins).

We thank the reviewer for this reasonable question. In our current work, we did not focus on the mechanisms that regulate permeability downstream of Tie2 but rather on acute (posttranslational) events that occur on the receptor level per se. However, we could show in vitro that VE-cadherin expression might be affected by Tie2 cleavage (Figure 3A). Others have shown that the anti-permeability effect of Tie2 is mostly mediated via changes in small GTPases (predominately RhoA) (3). Downstream signaling was beyond the scope of this project.

4) Recently, a study by Braun et al., 2020, from the group of Vestweber was published and reported on the fact that Tie2 activation limits the leakage that is induced by transmigrating leukocytes. This study would be complementary to theirs and should be cited in the Discussion section.

We thank the reviewer for this thoughtful advice that has been addressed and added to the paper.

Discussion:

“Further in this regard, a recent study by Braun et al. (Braun et al., 2020) reported that endothelial Tie2 activation limits diapedeses induced leakage.”

5) How sure can you be there are no cleavage sites outside of the synthetic Tie2 sites you generated? I am not familiar with the reliability of the software used, but apparently it at least wrongly predicted a cleavage site in D740-L. If the authors aren't sure these are the only three cleavage sites, perhaps the conclusion should be changed to leave open the possibility that MMP14 cleaves elsewhere (subsection “MMP14-dependent Tie2 cleavage occurs at the Fibronectin type III domain”).

We thank the reviewer for another interesting point raised. We agree that we cannot exclude the possibility of other unidentified cleavage sites within the Tie2 ectodomain. However, we believe that the observed 90% reduction of Tie2 cleavage by site-directed mutagenesis makes a strong case that these sites are indeed relevant. Nevertheless, as suggested we have made necessary corrections in the revised manuscript to acknowledge this important issue.

Results:

“In summary, this demonstrates that Tie2 is cleaved by MMP14 possibly at three different sites and…”

6) In the in vitro experiments, TNFa was used to induce inflammation, whereas in the in vivo experiments LPS was used. I understand that LPS is not as effective as TNFa in HUVEC, but perhaps the authors could explain possible other reasons for this.

In a previous study focusing on the molecular mechanisms that regulate Tie2 (4), in a pilot experiment, we tested various inflammatory mediators (including LPS) on ECs to replicate the loss of Tie2 observed in murine sepsis models. There, we found TNFα to be the most reproducible mediator of Tie2 changes in vitro. Consequently, we used TNF analogously for this study. Nevertheless, based on your suggestion we have repeated a key experiment to analyze potential mediator-associated differences. Author response image 2 shows similar potency with regard to Tie2 shedding (***P<0.001 Compared with control in Turkey`s multiple comparisons test. n=4 per group).

7) In Figure 3A, the holes between cells look really large. Is this image a representative image of all experiments? If so, maybe the authors can think of a way to quantify average hole size and/or number of holes induced by TNF-α.

We thank the reviewer for his/her suggestion. Yes, this is a representative image as indicated in the manuscript. The numbers of holes induced by TNFα are represented by the white arrows. We believe the HRP permeability experiment in Figure 3C is a better quantitative approach to measure the amount of permeability. Nevertheless, based on your suggestion we have quantified the holes. Figure 3B shows the average gap/hole number induced by TNFα (*P<0.05, **P<0.01, ***P<0.001 Compared with control in Turkey’s multiple comparisons test. n=3 per group).

References:

1) I. Kim, S. O. Moon, S. K. Park, S. W. Chae, G. Y. Koh, Angiopoietin-1 reduces VEGF-stimulated leukocyte adhesion to endothelial cells by reducing ICAM-1, VCAM-1, and E-selectin expression. Circ Res 89, 477-479 (2001).2) A. Tadros, D. P. Hughes, B. J. Dunmore, N. P. J. Brindle, ABIN-2 protects endothelial cells from death and has a role in the antiapoptotic effect of angiopoietin-1. Blood 102, 4407-4409 (2003).3) T. Mammoto et al., Angiopoietin-1 requires p190 RhoGAP to protect against vascular leakage in vivo. Journal of Biological Chemistry 282, 23910-23918 (2007).4) K. Thamm et al., Molecular Regulation of Acute Tie2 Suppression in Sepsis. Crit Care Med 46, e928-e936 (2018).